# Naphthalene Diimides Carrying Two β-Cyclodextrins Prefer Telomere RNA G-Quadruplex Recognition

**DOI:** 10.3390/molecules27134053

**Published:** 2022-06-23

**Authors:** Tingting Zou, Yuka Sato, Shuma Kaneyoshi, Kota Mano, Rui Yasukawa, Yoshifumi Nakano, Satoshi Fujii, Shinobu Sato, Shigeori Takenaka

**Affiliations:** 1Department of Applied Chemistry, Kyushu Institute of Technology, Fukuoka 804-8550, Japan; zoutt19880808@gmail.com (T.Z.); ysato@takenaka.che.kyutech.ac.jp (Y.S.); skaneyoshi@takenaka.che.kyutech.ac.jp (S.K.); kmano@takenaka.che.kyutech.ac.jp (K.M.); ryasukawa@takenaka.che.kyutech.ac.jp (R.Y.); ynakano@takenaka.che.kyutech.ac.jp (Y.N.); 2Department of Bioscience and Bioinformatics, Kyushu Institute of Technology, Fukuoka 820-8502, Japan; sfujii@bio.kyutech.ac.jp

**Keywords:** naphthalene diimide, β-cyclodextrins (β-CyDs), parallel G-quadruplex, DNA, RNA, nucleobase inclusion, apyromidinic sites, c-*myc*, c-*kit*, telomere RNA

## Abstract

Newly synthesized naphthalene diimide carrying two β-cyclodextrins (NDI-β-CyDs) showed improved specificity for the parallel G-quadruplex structure alongside the hybrid G-quadruplex structure. Specifically, the highest binding affinity of NDI-β-CyDs for the telomere RNA G-quadruplex was observed. The binding simulation indicated that β-cyclodextrins might be available for loop nucleobase inclusion under its complex.

## 1. Introduction

G-quadruplex structures that comprehensively exist in the gene promoter region have been reported, which might regulate gene transcription and expression, and in the telomere region that is involved in inhibiting telomerase’s activity, further inducing cancer cell death [1,2,3]. Therefore, G-quadruplex-specific targeting has been considered a promising anticancer therapy with less side effects due to its biological roles [4,5]. Within the progress of NMR and crystal structure analysis techniques, plenty of G-quadruplex structures with varied sequences have been elucidated in detail [6]. Generally parallel, antiparallel, and hybrid (mixed type) structures were considered as the classical G-quadruplex forms [7]. The telomere DNA G-quadruplex can form a hybrid G-quadruplex with a lateral loop and diagonal loop [8], while the c-*myc* promoter region forms a flat parallel G-quadruplex with site chain loops [9].

Small molecules that selectively recognize the G-quadruplex structure, also termed as G4 ligands, show high potential for anticancer applications [10]. A large number of G4 ligands for G-quadruplex binding have been elucidated with DNA sequencing in recent years [11,12,13,14,15,16,17,18,19,20,21,22,23,24,25,26,27], while increasing evidence suggests that high-affinity G4-binding small molecules may exhibit a preference for binding to multiple G4s with a parallel topology [15], RNA G4s, which are more likely to form a parallel G-quadruplex structure with higher stability, and are gaining more attention as attractive targets for G4 ligands [16].

In particular, since it is known that the inhibition of transcription of telomeric repeat-containing RNA (TERRA), as a telomeric RNA [28], causes the inhibition of telomere length maintenance or ALT activity, pyridostatin [27] and naphthalene diimide derivatives [14,15,17,18,19,20,21,22,23,24,25,26] have been developed as ligands that bind to TERRA.

However, it is well known that telomere RNA G-quadruplex forms a parallel structure from Circular Dichroism (CD) spectra identification, and a parallel G-quadruplex was detailed and identified within two 12nt telomere RNA sequences [29]. There is still relatively little information about RNA G-quadruplexes, especially longer telomere RNA G-quadruplexes.

Due to the varied preference of molecules for different G-quadruplex topologies, G4 ligands, which can discriminate G-quadruplex patterns, are considered promising G4 ligands with better selectivity. Some promising strategies have been developed for discriminating G-quadruplexes and dsDNA, such as the use of naphthalene diimide, which was designed to interfere with its binding to dsDNA, and therefore enhance its selectivity towards the G-quadruplex [17,18]. Some ligands prefer to recognize the lateral and/or diagonal loop of the hybrid G-quadruplex, and some conjugators even specifically insert into the TTA linker pocket of the two-telomere DNA G-quadruplex [30]. Here, we reported a new strategy that enhances the selectivity of the G4 ligand for the parallel G-quadruplex, especially of the parallel telomere RNA G-quadruplex.

As shown in Figure 1A, we conjugated two β-cyclodextrins with naphthalene diimide to obtain NDI-β-CyDs, **1** and **2**. The NDI plane provides recognition and stacking with the G-quartet [19,26], and the bulk body of β- cyclodextrin inhibits the interaction of NDI-β-CyDs with dsDNA due to steric hindrance, and also disturbs its binding to G-quadruplexes with side chain loops. Thus, a flat parallel G-quadruplex was proposed that would be preferable for binding (Figure 1B). To identify whether this concept is feasible or not, two NDI-β-CyDs, **1** and **2**, were synthesized with different lengths of the linker between NDI and β-CyDs, and **3** was adopted as a control ligand for comparison.

## 2. Results and Discussion

The synthesis and confirmation of **1** and **2** were listed in Appendix A, **3** was prepared according to the previous report [31].

### 2.1. Binding Behaviors of **1** and **2** with DNA or RNA G-Quadruplex

Classical G-quadruplex DNA and RNA sequences were adopted to conduct the investigation (Table 1). CD spectra of these sequences were determined by adding ligands; for all these DNA sequences, **3** showed influence with CD spectra shift (292 nm up shift of Telomere G4, 267 nm down shift of c-*myc* and c-*kit*, 282 nm up shift of ds oligo), **1** and **2** displayed less impact on telomere G4, c-*kit* and ds oligo, while similarly, a 267 nm shift of c-*myc*’s CD spectra was obtained, which suggested that **1** and **2** might prefer c-*myc* (forming parallel G-quadruplex structure) recognition (Appendix A). Although c-*kit* was also proposed, which forms a parallel G-quadruplex, it was identified with a unique structural scaffold with four loops that interfere with ligand recognition [31]. For telomere RNA G-quadruplexes (23nt, 12nt), adding **1** and **2** could induce the CD spectra to shift both the telomere RNA G-quadruplex, which further indicated the recognition and binding of **1** and **2** for the parallel G-quadruplex (Figure 2, Appendix A).

UV-Vis measurements were then performed at 25 °C when Telomere G1, c-*myc*, and ds-oligo were added to a solution of 50 mM Tris-HCl (pH 7.4), and 100 mM KCl containing 5 μM **1**–**3**, respectively (Figure 3 and Appendix A). The UV-Vis spectra were saturated when c-*myc* was added in **1** or when Telomere G1, c-*myc*, and ds-oligo were added in **3**. The binding constants are summarized in Table 2 and were obtained using a Scatchard analysis of these measured data.

In other cases, however, the change in absorbance with the addition of DNA was small and the UV-Vis spectrum was not saturated. In this case, the Benesi–Hildebrand analysis was performed (Appendix A). It was found that cyclodextrin-free **3** could bind to Telomere G1, c-*myc*, and ds-oligo, but the binding constant for c-*myc* was lower than that of cyclodextrin-containing **1** (8.6 × 10^5^ M^−1^). This indicates that cyclodextrin is highly selective for **1** to c-*myc*. Similarly, cyclodextrin-containing **2** showed higher affinity for c-*myc* (1.0 × 10^5^ M^−1^) than Telomere G1 and ds-oligo. Hypochromicity (%) was generally correlated with the binding constant, and when the binding constant was high, the hypochromicity was also large, suggesting good stacking between naphthalene diimide and the base.

To identify whether **1** and **2** showed improved specificity for parallel G-quadruplex binding, isothermal titration calorimetry (ITC) was adopted to evaluate the accurate binding properties (Figure 4 and Appendix A, Table 3). Some substituted naphthalene diimides were designed with enhanced selectivity and showed marked biological performance [20,21,22,25,26]. In particular, sugar-modified trisubstituted naphthalene diimides showed high G-quartet DNA and RNA selectivity [25], while the naphthalene diimide plane could bind to the G-quadruplex structure with no discrimination of different sequences, and demonstrated a comparable affinity for interacting with ds-oligo, which limited the application of the unmodified naphthalene diimide plane as a G4-specific ligand. Similarly, the newly constructed naphthalene diimide substitutions **1** and **2** barely bound to ds-oligo (undetectable in ITC measurement, and *K*_a_ was calculated using UV-Vis with Scatchard plot analysis), and both showed a decreased response for the hybrid telomere G4 sequence, especially **2**, which was almost completely shut off from telomere G4 recognition. However, it is noticeable that although **1** and **2** were able to recognize the c-*myc* G-quadruplex, their binding affinity was weaker than **3** (Table 3).

In contrast, when testing the affinity of ligands for telomere RNA G-quadruplex formed by G1-r23nt, both NDI-β-CyDs showed a stronger affinity than **3**, especially **1**, which displayed the *K*_a_ for G1-r23nt at 19 × 10^5^ M^−1^.

The binding constant of **3** to G1-r23nt was lower than that to Telomere G1. The affinity of **3** for c-*myc*, a parallel structure, was also somewhat lower than for Telomere G1, a hybrid structure. Nitrogen-bearing 2-substituted NDI is known to hydrogen bond to the phosphate in the lateral loop of telomere G1 [26]. When **3** is stacked in the G quartet, it can hydrogen bond to the lateral loop in Telomere G1 where the lateral loop is present, but in the parallel structure, hydrogen bonding is not effective and affinity is expected to be slightly lower.

A 12nt parallel telomere RNA G-quadruplex (G1-r12nt) which has been reported with structure details was also adopted for investigation [17]. When compared to G1-r23nt, both NDI-β-CDs displayed a weaker affinity for the intramolecular parallel RNA G-quadruplex. Based on this result, the telomere RNA G-quadruplex (G1-r23nt) formed a distinct G-quadruplex structure with G1-r12nt, with an unconnected 12U, 1U, and 2A structure, especially the **3** side chain of G1-r23nt, which might be more useful for the recognition of NDI-β-CyDs. Compared to the poor performance of **3** in discriminating ds-oligo or different G-quadruplex structures, both **1** and **2** showed an enhanced affinity for the G-quadruplex. Furthermore, **1** and **2** could prefer recognizing G1-r23nt by more than 10 fold higher than for telomere G4 (10.7 folds of **1**, and 20.5 folds of **2**), which could be considered superior specificity for targeting the telomere RNA G-quadruplex. Regarding the structure, the 2′-OH hydroxyl groups in the RNA quadruplex play a significant role in redefining the hydration structure, which may provide a positive environment for NDI-β-CyDs with β-CyD as a side chain to interact with the loops [26].

Compared to **3**, the binding of **1** and **2** with the G-quadruplex generally produced larger negative ΔS values, which indicated a more stable condition of the NDI-β-CyDs—G-quadruplex complex, and the ΔS value of **1** with G-quadruplex was more negative than **2**, which also suggested that **1****,** with a shorted linker between NDI and β-CyD, may restrict the adjustability more during the binding process. However, except for G1-r23nt, similar thermal parameter values but distinct *K*_a_ values among **1**, **2** and **3** were obtained (Table 3).

### 2.2. Binding Model of **1** and **2** with DNA or RNA G-Quadruplex

The binding model of NDI-β-CyDs with G-quadruplex structures was simulated by computer calculation (Figure 5 and Appendix A). Other than the difficulty observed in fitting **1** and **2** with the hybrid telomere DNA G-quadruplex (unable to get a stable model), for both c-*myc* and the G1-r12nt parallel G-quadruplex structure, stable binding models were successfully simulated. Interestingly, in both models, except for the stacking of the NDI plane with G-quartet, β-CyDs were observed as a pocket for nucleobase inclusion (T7 and T16 in c-*myc*; U7 and U18 in G1-r12nt). Since previous reports supported the interaction between β-CyD and nucleobase [32], the c-*myc* sequence within two AP sites (apyromidinic sites without the base of T7 and T16) was adopted to investigate its influence on NDI-β-CyDs’ recognition of the G-quadruplex. Although a slight decreasing affinity of **1** or **2** for the c-*myc* AP site was observed with increased entropy (Figure 6, Table 3), it is still difficult to directly claim whether such nucleobase inclusion could indeed occur during G4 ligand binding with the G-quadruplex or not. Moreover, varied positioning of β-CyDs for c-*myc* or G1-r12nt was observed, which indicated that the shape of the G-quadruplex could be an important factor impacting NDI-β-CyDs’ G-quadruplex recognition.

Finally, we analyzed the stabilization effect of **1** and **2** for the G-quadruplex structure based on T_m_ measurement (Figure 7, Table 4). A comparison of **3**, **1** and **2** revealed a weaker ability to enhance the thermal stability of the G-quadruplex, which might be because the bulky structure of β-CyDs causes more resistance to maintain the NDI-β-CyDs-G-quadruplex complex, while there was almost no stabilization of NDI-β-CyDs for the telomere DNA G-quadruplex, and a comparatively stronger ability of NDI-β-CyDs for stabilizing the telomere RNA G-quadruplex was observed.

## 3. Materials and Methods

### 3.1. Generals

DNA and RNA oligonucleotides (Table 1) were purchased from Hokkaido System Science Co., Ltd. (Sapporo, Japan). Before use, the DNA was annealed under the following conditions: heating to 95 °C for 10 min, then cooling to 25 °C at 0.5 °C/min. A previously reported procedure was used to synthesize 3 [33].

UV/Vis and CD spectra were measured with 50 mM Tris-HCl, 100 mM KCl to fix the salt concentration. However, the ITC measurement was difficult with 50 mM Tris-HCl buffer (pH 7.4) containing 100 mM KCl, and the measurement conditions were the recommended conditions of the nano ITC (50 mM KH_2_PO_4_-K_2_HPO_4_ buffer, pH 7.0). Before performing the ITC measurement, the CD spectrum measurement of G4 with the addition of KCl was performed in a 50 mM KH_2_PO_4_-K_2_HPO_4_ buffer (pH 7.0), following which it was confirmed that the spectrum of the G4 structure did not change and that a G-quadruplex structure was sufficiently formed. In addition, when Tm a measurement could not be performed with 50 mM Tris-HCl, 100 mM KCl, the conditions were changed. Since the Tm curves of G1-r23nt and c-*myc* at 100 mM KCl did not change completely, the KCl concentration was set to 5 mM. On the contrary, G1-r12 was insufficient for the Tm measurement at a KCl of 100 mM, so it was measured with 50 mM KH_2_PO_4_-K_2_HPO_4_ buffer (pH 7.0) and 100 mM KCl.

The ^1^H NMR spectra were recorded using an AVANCE400 Spectrometer (Brucker Co.) operating at 400 MHz with tetramethylsilane (TMS) as the internal standard. Reversed phase High Performance Liquid Chromatography (HPLC), equipped with a SHIMADZU SCL-10A system controller, LC-20AD prominence LIQUID CHROMATOGRAPH, DGU-20A3 prominence degasser, SPD-M20A prominence DIODE ARRAY DETECTOR, and CTO-20A prominence COLUMN OVEN (Shimadzu Co., Kyoto, Japan), was carried out in the gradient mode. The concentration of solution B was changed (from 10% to 100%, 30 min) at 40 °C (Solution A: 0.1% trifluoroacetic acid, Solution B: 70% acetonitrile, 0.1% trifluoroacetic acid, Flow rate: 1 mL/min, Column: Inertsil ODS-4 (GL Sciences Inc., Tokyo, Japan)). High-resolution mass (HRMS) spectra were obtained with a JEOL JMS-SX102A mass spectrometer (JEOL Ltd., Tokyo, Japan) with an electron ionization (EI) method with a double-focusing sector detector. The mass spectra were recorded using a Microflex (Bruker Co., Billerica, MS, USA) in the time-of-flight mode with Dihydroxybenzoic acid (DHBA) as the matrix.

### 3.2. Synthesis of ***1***

*N*,*N*′-Bis(3-methylaminopropyl)naphthalene-1,4,5,8-tetracarboxylic acid diimide and Tosyl-β-cyclodextrin (CyD) were synthesized by the route previously reported [34,35]. See Figure 1.

*N*, *N*′-Bis(3-methylaminopropyl)naphthalene-1,4,5,8-tetracarboxylic acid diimide 0.2 g (0.5 mmol) and Tosyl-β-CyD 2.7 g (2.1 mmol) were dissolved in 10 mL, and mixed in 70 °C for 48 h. Then, the solvent was removed by reduced pressure evaporation, and 10 mL of 0.1% trifluoroacetic acid (TFA) containing 7% acetonitrile was added and mixed in room temperature, after which the solution was collected by filtration. The target compound **1** was purified by RP-HPLC, and dried by freeze drying to obtain white powder. **1** was confirmed by ^1^H-NMR (Appendix A), HPLC (Appendix A) and HRMS (EI^+^).

^1^H-NMR (400 MHz, D_2_O): δ = 2.10–2.20 (4H, N(CH_3_)CH_2_C*H*_2_CH_2_, br), 2.74–8.85 (6H, N(C*H*_3_), m), 3.16–4.29 (96H, C2*H*, C3*H*, C4*H*, C5*H*, C6*H* of β-CD, and CD-C*H*_2_N(CH_3_)C*H*_2_CH_2_C*H*_2_), 4.95–5.13 (14H, C1*H* of CD, m), and 8.70–9.14 (4H, Ar*H*, m) ppm. HRMS (EI+) *m*/*z* [M]+ Calcd for C_106_H_160_N_4_O_72_H: 2641.90597, found 2641.89364. Yield 62 mg, yield ratio 48%.

### 3.3. Synthesis of ***2***

A total of 50 mL was added to 1,4,5,8-Naphthalenetetracarboxylic acid 1.1 g (4.1 mmol) and Glycine 1.3 g (18 mmol), and reflux was performed with heating for 6 days. Then, the solution was cooled to room temperature, and kept on ice for 1 h. The yellow precipitation was collected by suction filtration, and rinsed with MilliQ several times. After reduced pressure drying, the obtained solid was washed with 100 mL acetonitrile, and the precipitation was dried under reduced pressure drying again. **4** was obtained and confirmed with MALDI-TOF-MS (Appendix A), and ^1^H-NMR (Appendix A). Yield 1.5 g (3.8 mmol), yield ratio 94%. ^1^H-NMR (400 MHz, DMSO): δ = 4.76 (4H, NC*H*_2_CO, s), 8.70 (4H, Ar*H*, d), and 13.23 (2H, COO*H*, br) ppm. See Figure 2:

A total of 1.0 g (0.77 mmol) of Tosyl-β-CyD was dissolved by *N*-methylpyrrolidone 2.5 mL under ultra-sonication. After complete dissolving, diethylamine 134 µL (2.0 mmol) and one grain of potassium iodide were added to the solution and mixed in an 80 °C oil bath for 7 h. After mixing, the solution was cooled down to room temperature, 70 mL of ethanol was added, and the obtained precipitation was collected by suction filtration and washed by methanol and diethyl ether, then dried under reduced pressure drying to obtain a light yellow solid. The obtained target compound **5** was confirmed by TLC (1-butanol: ethanol: H_2_O = 5:4:3) using the *p*-Anisaldehyde Stain and Ninhydrin test.

A total of 0.04 g (0.1 mmol) of 4, 0.44 g (0.36 mmol) of **5**, 1-[Bis(dimethylamino)methylene]-1*H*-1,2,3-triazolo [4,5-*b*]pyridinium 3-Oxide Hexafluorophosphate (HATU) 0.096 g, and triethylamine 42 μL were added to 5 mL of *N*,*N*-dimethylformamide and dissolved by ultra-sonication, then mixed for 90 h. The solvent was removed by reduced pressure evaporation. Then, target compound **2** was purified by RP-HPLC (retention time around 10 min), and dried under frozen drying. **2** was obtained as a white solid, and confirmed by ^1^H-NMR (Appendix A), HPLC (Appendix A) and HRMS (EI^+^). ^1^H-NMR (400 MHz, D_2_O): δ = 3.02–3.86 (96H, C2*H*, C3*H*, C4*H*, C5*H*, C6*H* of β-CD, and N-C*H*_2_CONHC*H*_2_C*H*_2_NHC*H*_2_, m), 4.82–4.94 (14H, C1 of CD, m), 7.51–7.55 (2H, CH_2_CON*H*CH_2_, m) and 8.28–8.93 (4H, ArH, 4H) ppm. HRMS (EI^+^) *m*/*z* [M]^+^ Calcd for C_106_H_158_N_6_O_74_Na: 2721.86824, found 2721.85922. Yield 103 mg, yield ratio 37%.

### 3.4. Circular Dichroism (CD) Measurements

CD spectra of the annealed 1.5 μM DNA or RNA were measured in 50 mM Tris-HCl buffer (pH 7.4) with 100 mM KCl at 25 °C, with a JASCO J-820 spectrophotometer equipped with a temperature controller, in the presence of 0 μM to 4.5 μM of **1**–**3**. The measurement was performed at a scan rate of 50 nm/min, using a Jasco J-820 spectropolarimeter (Tokyo, Japan) with the following conditions: response, 4 s; data interval, 0.2 nm; sensitivity, 100 mdeg; bandwidth, 2 nm; and scan number, 4 times.

### 3.5. Isothermal Titration Calorimetry (ITC) Measurements

ITC measurements were performed using a low volume nano ITC (TA instruments, USA) with a cell volume of 190 μL at 25 °C. The annealed DNA solution and chemicals were degassed for 10 min before loading. The measurement was performed with titrating **1**, **2** or **3** (0–100 μM) to the G-quadruplex (telomere G1, c-*myc*, c-*myc* AP site, G1-r23nt, G1-r12nt) in 50 mM KH_2_PO_4_-K_2_HPO_4_ buffer (pH 7.0) at 25 °C. In each titration, 2 µL of the ligand solution was injected into a quadruplex solution every 120 s up to a total of 25 injections, using a computer-controlled 50 µL microsyringe, with stirring at 300 rpm. The binding curve was fitting on the independent binding model.

### 3.6. UV-Vis Absorption Spectroscopy

The binding affinity of **1**–**3** to telomere G1, c-*myc* and ds-oligo were studied with Hitachi U-3310 spectrophotometer (Tokyo, Japan). A total of 120 μL of 150 μM telomere G1, c-*myc* and ds-oligo were added to 5 μM of **1**, **2** or **3** in 50 mM Tris-HCl buffer (pH 7.4) and 100 mM KCl. Absorbance spectra were taken at 25 °C. The observed spectrum changes at 385 nm were rearranged with a Scatchard plot with the following equation in the case of non-cooperative binding,
ν/L = K(n − ν), (1)
where ν, L, n, and K refer to the saturation fraction as the amount of bound ligand per added DNA, amount of unbound ligand, binding number of ligands per one G4, or double stranded DNA, and binding affinity, respectively. When the absorption spectrum was not saturated, nK values were obtained using the Benesi–Hildebrand method and the following Equation [36],
1/ΔAbs = 1/(*l*Δε [ligad]) + 1/(nK*l*Δε [ligad]) × (1/DNA)(2)

### 3.7. Apparent Melting Temperature (Tm) Detection

Melting curves of annealed 1.5 μM DNA or RNA were measured in 50 mM Tris-HCl buffer (pH 7.4) and 100 mM KCl for Telomere G1 (288 nm); 50 mM Tris-HCl buffer (pH 7.4) and 5 mM KCl for c-*myc* (264 nm); G1-r23nt (263 nm); 50 mM KH_2_PO_4_-K_2_HPO_4_ buffer (pH 7.0) and 100 mM KCl for G1-r12nt (263 nm) in the absence or presence of 4.5 μM of **1**–**3**. The measurement was conducted using a Jasco J-820 spectrophotometer equipped with a temperature controller with the following conditions: response; 100 mdeg, temperature gradient; 60 °C/h, response; 1 s; data collecting interval; 0.5 °C, and bandwidth; 1 nm. Overall, 3 ml of the total volume was used in the cell with 1 cm of light path length.

### 3.8. Modeling Simulations

Molecular Modeling of these complexes was constructed with MOE [37]. Data of NMR in the aqueous solution of K^+^ (PDBID: 1xav (c-myc); 2kbp (G1-r12nt)) were utilized in the structural construction of the binding model simulation. The AMBER10:ETH [38,39] force field was used for this modeling simulation. The following modeling simulation processes were performed with the tether constrained to the three planes of the G-tetrad core in the structure. **1** or **2** was placed on the binding site of Telomere G1 and the energy minimization of these complexes was carried out in order to resolve the 3D obstacle. As a next step, the models interacting between β-CyDs and nucleobase (T7 and T16 in 1xav; U6 and U18 in 2kbp) were created. Molecular dynamics simulations of these complexes with distance restraints between three atoms of the β-CyDs and three atoms of the nucleobases were carried out until **1** or **2** was located in the binding site as a stable condition. Additionally, the distance restraints were then removed from the complexes, and molecular dynamics simulations for 500 ps were carried out. The complexes were observed to maintain the binding form in the simulation. Finally, the complex molecular models by energy minimization were obtained as shown in Figure 5 and Appendix A.

## 4. Conclusions

Newly constructed NDI-β-CyDs, **1** and **2****,** were identified with significant specificity for recognizing parallel G-quadruplex, especially for a 23nt telomere RNA G-quadruplex. β-CyD might be an interesting medium to enhance the ability to perform this discrimination, whereas due to the bulky structure of β-CyD, the stabilization effect of NDI-β-CyDs for the G-quadruplex weakened. Some β-CyDs analogs with smaller volumes might be good candidates when conjugating with NDI to improve their stabilization performance.

## Data Availability

Data are contained within the article or Appendix A.

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
