# Peer review of "Naphthalene Diimides Carrying Two β-Cyclodextrins Prefer Telomere RNA G-Quadruplex Recognition"

_molecules, 2022, doi:10.3390/molecules27134053_

Round 1
Reviewer 1 Report
The manuscript entitled “Naphthalene diimides carrying two β-cyclodextrins prefer Telomere RNA G-quadruplex recognition” has described the synthesis of two newly G4 binding ligands, naphthalene diimide carrying two β-cyclodextrins (NDI-β-CyDs). The authors demonstrated the incorporation of β-cyclodextrins could improve the binding selectivity of NDI-β-CyDs toward parallel RNA G4 rather than hybrid telomeric DNA G4. The experiments are well designed and the conclusion is favored by the experimental data. Several important queries/suggestions needed to be addressed as follows:
1. The reviewer is confused by this statement “the bulk body of β-cyclodextrin inhibits the interaction of NDI-β-CyDs with dsDNA due to steric hindrance, and also disturbing its binding to G-quadruplex with out-loop. Thus, a flat parallel G-quadruplex was supposed that would be preferable for the specific binding.” The two RNA G4 sequences used in this study also have two (12-mer) or three (23-mer) side chain loops, respectively, which may show a steric hindrance effect to β-cyclodextrin.
2. Figure 4, thermodynamic parameter of c-myc AP site and c-myc with 1-3 should be listed in a table, consistent with Tables 2 and 3.
3. The manuscript needs to be further edited and checked.
Author Response
Correspondence
Thank you very much for your detailed comments. These comments are very helpful to improve the level of our paper. I will address the individual comments below.
Reviewer#1
The manuscript entitled “Naphthalene diimides carrying two β-cyclodextrins prefer Telomere RNA G-quadruplex recognition” has described the synthesis of two newly G4 binding ligands, naphthalene diimide carrying two β-cyclodextrins (NDI-β-CyDs). The authors demonstrated the incorporation of β-cyclodextrins could improve the binding selectivity of NDI-β-CyDs toward parallel RNA G4 rather than hybrid telomeric DNA G4. The experiments are well designed and the conclusion is favored by the experimental data. Several important queries/suggestions needed to be addressed as follows:
- The reviewer is confused by this statement “the bulk body of β-cyclodextrin inhibits the interaction of NDI-β-CyDs with dsDNA due to steric hindrance, and also disturbing its binding to G-quadruplex with out-loop. Thus, a flat parallel G-quadruplex was supposed that would be preferable for the specific binding.” The two RNA G4 sequences used in this study also have two (12-mer) or three (23-mer) side chain loops, respectively, which may show a steric hindrance effect to β-cyclodextrin.
Ans. The side chain loop of Parallel G-quadruplex may be a potential site of steric hindrance between NDI-b-CD and Hybrid type G4, since the side chain loop of Parallel G-quadruplex may encapsulate the base of NDI-b-CD. In this manuscript, we changed the word "loop" to "lateral loop," "diagonal loop," and "side chain loop" to make it easier for the Readers to understand.
Figure 4, thermodynamic parameter of c-myc AP site and c-myc with 1-3 should be listed in a table, consistent with Tables 2 and 3.
Ans. The results in c-myc AP site and c-myc described in Table 2 and 3 are summarized in Table 3.
The manuscript needs to be further edited and checked.
Ans. We did further edited and checked in the manuscript.
Reviewer 2 Report
The manuscript of Takenaka and Coll. describes the studies of novel naphthalene diimide derivatives as potential G-quadruplex ligands. They have attached a bulky cyclodextrin moiety at both side of the naphthalene aromatic ring to prevent the interaction with double-stranded DNA. Circular dichroism, thermal melting and ITC assays as well as modeling have been used to study the binding affinity with dsDNA and various G-quadruplexes.
The work is well carried out and the results are interesting. The manuscript should be accepted for publication in Molecules after minor revisions to be performed.
1/ The control compound 3 is less affine for G4 RNA that for G4 DNA. The authors should give an explanation for that.
2/ The authors have used a parallel RNA telomeric sequence (G1-r12nt) and observed that their diimide derivatives have less affinity with it in comparison with the other RNA sequence (G1-r23nt). They argued that it could be due to a difference of structure: it should be interesting for the reader to have the CD spectrum of this G1-r12nt and also with addition of compounds 1-3. I suppose that G1-r12nt forms an intermolecular parallel RNA G4 (not intramolecular as it is noted in page 3, line 103)?
3/ The legend for Figure S7 in the SI is missing: it seems that there are 4 different colors for the different curves but no information are given.
4/ The figures S1 and S4 (NMR spectra) are of poor quality. The authors should afford more visible figures for increase the clarity.
5/ It would be interesting to use SPR method for studying the interaction because it could afford the kinetics of association which, I suppose, is a crucial parameter for the interaction of these bulky derivatives.
In addition, the manuscript should be read carefully to correct some typological errors and/or English sentences phrasing.
Author Response
Correspondence
Thank you very much for your detailed comments. These comments are very helpful to improve the level of our paper. I will address the individual comments below.
Reviewer#2
1/ The control compound 3 is less affine for G4 RNA that for G4 DNA. The authors should give an explanation for that.
Ans. The paper (J. Phys. Chem. B 2015, 119, 3335−3347) suggested that the hydrogen of the 2-substituted NDI is hydrogen bonded to the Latelal loop of Telomere G1. 3 is similarly expected to be hydrogen bonded to Telomere G1. In the parallel G4 structure, this effect is less likely, and the binding constant is expected to be slightly lower.
MS 4P
The binding constant of 3 to G1-r23nt was lower than that to Telomere G1. The affinity of 3 for c-myc, a parallel structure, was also somewhat lower than for Telomere G1, a hybrid structure. Nitrogen-bearing 2-substituted NDI is known to hydrogen bond to the phosphate in the lateral loop of telomere G1 [26]. When 3 is stacked in the G quartet, it can hydrogen bond to the lateral loop in Telomere G1 where the lateral loop is present, but in the parallel structure, hydrogen bonding is not effective and affinity is expected to be slightly lower.
2/ The authors have used a parallel RNA telomeric sequence (G1-r12nt) and observed that their diimide derivatives have less affinity with it in comparison with the other RNA sequence (G1-r23nt). They argued that it could be due to a difference of structure: it should be interesting for the reader to have the CD spectrum of this G1-r12nt and also with addition of compounds 1-3. I suppose that G1-r12nt forms an intermolecular parallel RNA G4 (not intramolecular as it is noted in page 3, line 103)?
Ans. CD data for G1-r12nt added in this manuscript. [Figure 2(F), Figure S7(F), Figure S8(F)]
3/ The legend for Figure S7 in the SI is missing: it seems that there are 4 different colors for the different curves but no information are given.
Ans. A caption has been added to the figure. [Figure 2, Figure S7, Figure S8]
4/ The figures S1 and S4 (NMR spectra) are of poor quality. The authors should afford more visible figures for increase the clarity.
Ans. We prepared a higher resolution version.
5/ It would be interesting to use SPR method for studying the interaction because it could afford the kinetics of association which, I suppose, is a crucial parameter for the interaction of these bulky derivatives.
Ans. Since the equipment is not available, it is difficult to conduct this experiment at this time. However, we believe that our claims are suggested by the data discussed here.
In addition, the manuscript should be read carefully to correct some typological errors and/or English sentences phrasing.
Ans. We did correction carefully in our manuscript.
Reviewer 3 Report
The article of Zuo et al. deals with the study naphthalene diimide carrying two β-cyclodextrins as G-quadruplex binders. They performed circular dichroism and UV-vis absorption spectroscopy measurements, ITC experiments, and molecular modeling. Comments follow:
- The Introduction is not exhaustive. A lot of relevant recent literature concerning naphthalene diimide compounds (e.g., Eur. J. Med. Chem. 2022, 232, 114183, doi: 10.1016/j.ejmech.2022.114183; Int. J. Biol. Macromol. 2021, 166, 1320-34, doi: 10.1016/j.ijbiomac.2020.11.013) and telomere RNA (TERRA) G-quadruplex (e.g., Nat. Commun. 2021, 12, 3760, doi: 10.1038/s41467-021-24097-6; Int. J. Mol. Sci. 2021, 22, 10315, doi: 10.3390/ijms221910315) is not cited.
- The binding affinity of compounds 1-3 to ds-oligo was studied by UV-vis absorption spectroscopy, but the UV spectra and Scatchard plots are never shown either in the manuscript or in the supporting material.
- The authors compare the results obtained by UV-Vis Scatchard plot analysis with those obtained by ITC since some interactions (1 and 2 vs. ds-oligo) are undetectable in ITC measurements (Figure 2). They should also perform UV-Vis Scatchard plot analysis for compounds 1 and 2 with the investigated G-quadruplexes and compare the results.
- No spectra or ITC titrations are shown throughout the manuscript. The authors only show bar charts and tables (often containing the same information). They should completely change figures and tables.
- Thermal melting experiments were performed, but once again they are not shown in the manuscript.
- The information relating to molecular modeling in the Materials and Methods section is extremely poor and not detailed. This not only makes these calculations unreproducible but does not allow one to understand what has really been done.
- Some of the ITC results seem misinterpreted. For example, it appears that compound 2 is not able to bind to telomere G1, as well as compound 3 is not able to bind to r12nt. In addition, many of the stoichiometries appear to be incorrect (see Tables 2 and 3 vs. Figure S8).
Overall, this manuscript presents several serious concerns and doesn’t deserve publication in Molecules.
Author Response
Correspondence
Thank you very much for your detailed comments. These comments are very helpful to improve the level of our paper. I will address the individual comments below.
Reviewer#3
The article of Zuo et al. deals with the study naphthalene diimide carrying two β-cyclodextrins as G-quadruplex binders. They performed circular dichroism and UV-vis absorption spectroscopy measurements, ITC experiments, and molecular modeling. Comments follow:
- The Introduction is not exhaustive. A lot of relevant recent literature concerning naphthalene diimide compounds (e.g., Eur. J. Med. Chem. 2022, 232, 114183, doi: 10.1016/j.ejmech.2022.114183; Int. J. Biol. Macromol. 2021, 166, 1320-34, doi: 10.1016/j.ijbiomac.2020.11.013) and telomere RNA (TERRA) G-quadruplex (e.g., Nat. Commun. 2021, 12, 3760, doi: 10.1038/s41467-021-24097-6; Int. J. Mol. Sci. 2021, 22, 10315, doi: 10.3390/ijms221910315) is not cited.
Ans. The papers pointed out by the referee have been appended and cited as 24, 25, 27, and 28, respectively.
- The binding affinity of compounds 1-3 to ds-oligo was studied by UV-vis absorption spectroscopy, but the UV spectra and Scatchard plots are never shown either in the manuscript or in the supporting material.
Ans. The UV spectra of 1-3 ds-oligo have been added to Figure 3 and Supporting Material, Figure S9.
- The authors compare the results obtained by UV-Vis Scatchard plot analysis with those obtained by ITC since some interactions (1 and 2 vs. ds-oligo) are undetectable in ITC measurements (Figure 2). They should also perform UV-Vis Scatchard plot analysis for compounds 1 and 2 with the investigated G-quadruplexes and compare the results.
Ans. UV measurements were carried out with Telomere G1, c-myc, and dsoligo. G1-r12nt, G1-r23nt, and c-myc AP were not sufficiently abundant for custom synthesis, and thus only ITC was measured for these oligonucleotides. Those results were added to Figure 3 and Supporting Material Figure S9. These results obtained were similar to the ITC results.
- No spectra or ITC titrations are shown throughout the manuscript. The authors only show bar charts and tables (often containing the same information). They should completely change figures and tables.
Ans. Representative spectra and ITC charts are shown in MS; CD, UV, ITC, and Tm charts are shown as Figures 2, 3, 4, and 7, respectively; Table has been revised.
- Thermal melting experiments were performed, but once again they are not shown in the manuscript.
Ans. A Tm chart is shown in the manuscript as Figure 7.
- The information relating to molecular modeling in the Materials and Methods section is extremely poor and not detailed. This not only makes these calculations unreproducible but does not allow one to understand what has really been done.
Ans. Molecular modeling in the Materials and Methods section was revised follows.
4.8 Modeling simulation
Molecular modeling of these complexes was constructed by MOE [36]. Data of NMR in the aqueous solution of K+ (PDBID: 1xav (c-myc); 2kbp (G1-r12nt)) was utilized in the structural construction of binding model simulation. AMBER10:ETH [37,38] force field was used for this modeling simulation. Following modeling simulation processes were performed with the tether constrain to the three planes of the G-tetrad core in the structure. 1 or 2 was placed on the binding site of Telomere G1 and energy minimization of these complexes was carried out in order to resolve the 3D obstacle. As a next step, the models interacting between β-CyDs and nucleobase (T7 and T16 in 1xav; U6 and U18 in 2kbp) were created. Molecular dynamics simulations of these complexes with distance restraints between three atoms of the β-CyDs and three atoms of the nucleobases were carried out until 1 or 2 was located in the binding site as stable condition. And then, the distance restraints were removed from the complexes, molecular dynamics simulations for 500 ps were carried out. The complexes were observed to keep the binding form in the simulation. Finally, the complex molecular models by energy minimization was obtained as shown in Figure 5 and Figure S12.
- Some of the ITC results seem misinterpreted. For example, it appears that compound 2 is not able to bind to telomere G1, as well as compound 3 is not able to bind to r12nt. In addition, many of the stoichiometries appear to be incorrect (see Tables 2 and 3 vs. Figure S8).
Ans. Those that could not be fitted were assumed to be unanalyzable and the Table was revised. The text was also revised accordingly.
We apologize for any errors or lack of explanation in the manuscript. We believe that the corrected manuscript has a good chance of being accepted for Molecules.
Round 2
Reviewer 3 Report
The manuscript of Zuo et al. has significantly improved compared to the previous version, however there are still several concerns to clarify/improve before recommending its publication:
1. It is not clear to this referee why Isothermal titration calorimetry (ITC), UV-vis absorption, and Circular Dichroism (spectra and melting) measurements have been performed in different solution conditions. The authors should at least explain this decision in the manuscript.
2. CD melting experiments for the different nucleic acid molecules have been performed in different buffers and ionic strength (and pH!): 50 mM Tris-HCl buffer (pH 7.4) and 100 mM KCl; 50 mM Tris-HCl buffer (pH 7.4) and 5 mM KCl; 50 mM KH2PO4-K2HPO4 buffer (pH 7.0) and 100 mM KCl. The ionic strength of the solution clearly influences the stability of the nucleic acid structures, their interactions, and the ability of the ligands to stabilize them. Therefore, the data obtained cannot be compared correctly. Authors should use the same ionic strength or at least explain in the manuscript why this is not possible.
Other points:
- “TERAA” (p. 1, l. 41) should be “TERRA”.
- Please improve the quality of Figure 4.
- The errors on the values listed in Table 2, on the stoichiometry in Table 3, and on the deltaTm values in Table 4 are missing.
- “4.4 Circular Dichroism (CD) measurement” should be “4.4 Circular Dichroism (CD) measurements”.
- Since the presence of thermodynamic equilibrium has not been verified, “melting temperatures” should be defined as “apparent melting temperatures.
- “Modeling simulation” should be “Modeling simulations”.
Author Response
Please see the attachment.
The revised parts were described in blue in the revised MS.
